

# Regional water footprint assessment for a semi-arid basin in India

Mukesh Kumar Mehla

Department of Soil and Water Engineering, College of Technology and Engineering, Maharana Pratap University of Agriculture and Technology, Udaipur, Rajasthan, India

## ABSTRACT

Water footprint assessment enables us to pinpoint the impacts and limitations of the current systems. Identifying vulnerabilities across various regions and times helps us prepare for suitable actions for improving water productivity and promoting sustainable water use. This study aims to provide a comprehensive evaluation of the sector-wise water footprint in the Banas River Basin from 2008–2020. The water footprint of the Banas River Basin was estimated as 20.2 billion cubic meters (BCM)/year from all sectors. The water footprint has increased over the year with the increase in population, the number of industries, and crop production demand. The average annual water footprint of crop production varied from 11.4–23.1 BCM/year (mean 19.3 BCM/year) during the study period. Results indicate that the water footprint has nearly doubled in the past decade. Wheat, bajra, maize, and rapeseed & mustard make up 67.4% of crop production's total average annual water footprint. Suitable measures should be implemented in the basin to improve water productivity and promote sustainable water use in agriculture, which accounts for nearly 95.5% of the total water footprint (WF) of the Banas basin. The outcomes of the study provide a reference point for further research and planning of appropriate actions to combat water scarcity challenges in the Banas basin.

## INTRODUCTION

India is the second most populous country in the world. It supports nearly 17.1% of the world's population (≈1.3 billion) and 20% of the world's livestock population (≈500 million), resulting in increased stress on limited freshwater resources (*Jain, 2019*). Efficient use of available water resources is vital for a nation like India, where the agriculture sector is the leading consumer of water. Over time average annual *per capita* water availability has declined from 1,816 in 2001 to 1,545 cubic meters in the year 2011, and it is projected to further go down to 1,486 cubic meters by the year 2021. It will be 1,367 cubic meters by 2031 (*PIB, 2020*). Water availability and allocation have become critical issues worldwide, particularly in arid and semi-arid regions. Water security is essential for social and economic development, enhancing health, well-being, and economic progress, particularly in developing countries (*Mekonnen & Hoekstra, 2013*). Nearly two-thirds of the world's population currently faces water scarcity for at least one month per year (*Mekonnen &*

Corresponding author
Mukesh Kumar Mehla,
mukeshmehla310@gmail.com

*Hoekstra, 2016*). Irrigation water use is essential, especially in the current scenario where water scarcity and climate change are becoming significant threats worldwide. The functionality of irrigation is not limited to providing sufficient water for crops to achieve better production outcomes (*Tesema et al., 2011*). Technology and management practices play an essential role in reducing inessential water use. Certain challenges are being posed by climate change, water scarcity, and growing demand from other sectors. Thus, promoting efficient and sustainable water use with better planning has become imperative (*Hoekstra, 2017*). There is a need to develop better water management policies to meet our current and future demands, ensuring food security and fulfilling domestic and industrial needs. Also, suitable measures should be taken to increase water use efficiency and reduce the water demands of agricultural production.

The water footprint (WF) is a broad concept that indicates water consumption within a region for a product, commodity, process, or service (*Hoekstra et al., 2009*). Calculated by summing the volume of direct and indirect water used for a product, commodity, process, or service. Several WF studies have been conducted worldwide at various scales (*Hoekstra, 2017*). Planning and managing water resources at the river basin scale is essential for increasing water availability and improving water quality while ensuring long-term sustainability. WF assessment helps understand the importance of sustainable water utilization and forms a basis for global freshwater management efforts (*Chukalla, Krol & Hoekstra, 2015*; *Mekonnen & Hoekstra, 2016*). Water scarcity assessment faces the challenges of incorporating green water, water quality, environmental flow requirements, globalization, and virtual water trade-related issues (*Liu et al., 2017*). Different crop models like Aqua crop, DSSAT, APSIM, and WOFOST (yield gap) have been used earlier to study the effect of soil moisture stress, deficit irrigation, nutrient stress, sowing date, and impact of climate change on crop growth and productivity (*Tenreiro et al., 2020*). Various factors affecting water use efficiency include poor agricultural practices, inefficient irrigation systems technology, and water pricing. Mitigating water scarcity has become a significant concern globally, and numerous studies have been conducted on this (*Wada, Wisser & Bierkens, 2014*; *Kummu et al., 2016*; *Zhuo et al., 2016*). WFs had been quantified at high spatial and temporal resolution (*Mekonnen & Hoekstra, 2011*, *2014*; *Hoekstra & Mekonnen, 2012*). Inter- and intra-annual variability of water availability and trends in WFs have been studied (*Zhuo et al., 2016*; *Liu et al., 2017*).

River basins have seen a decline in *per capita* water availability all over India due to continuous population pressures, agriculture, and industrial expansion (*Dhawan, 2017*). Freshwater availability for agricultural purposes in India is less than required owing to the high WF and poor farming practices (*Kampman, Hoekstra & Krol, 2008*). To ensure sustainability at a river basin scale, capping/limiting the consumptive and degradative water use per river basin was proposed so that water use stays within maximum sustainable levels (*Hoekstra, 2014*). At the river basin scale, WF analysis can address certain policy and water management-related issues to facilitate a more efficient allocation and use of water resources, providing a framework for policy formulation (*Mali et al., 2018*; *Nouri et al., 2019*; *Khan et al., 2021*). WF modelling enables us to pinpoint the impacts and limitations of the current crop production system. Assessing vulnerabilities across agricultural

management systems across various regions and times helps us prepare for suitable actions for improving water productivity and promoting sustainable water use.

The current literature provides crop WF for various areas worldwide and a global average for comparison, but most use global or national statistics. WF can vary significantly spatially and temporarily, even within the basin. Water allocation strategies and crop planning for efficient water use should be done considering a long-term perspective and local conditions. The findings from this paper will benefit the farmers and water resource planners in the basin. This research will also assist decision-makers in implementing proper agricultural governance and measures that will help in ensuring global food and water security without endangering the environment. Outcomes provide baseline information for further research and will provide imperative insights into the current situation in the basin. This will assist in planning appropriate measures to overcome water scarcity challenges and reduce the water footprint in the basin. This study integrates local data and robust modeling capabilities of the AquaCrop model to more precisely assess the WFs of major crops of the basin alongside estimates from other important sectors which are generally not considered. Considering all these points, this study was undertaken with the aim of evaluating the sector-wise water footprint in the Banas River Basin.

# MATERIALS AND METHODS

## Study area

The Banas River Basin (BRB) lies between 24°15′–27°20′ latitudes and 73°25′–77°00′ longitudes (Fig. 1). It has a catchment area of 47,060 km$^2$ (4.7 Mha) within Rajasthan (*WRD, 2014a*). This study aims to determine the sector-wise water footprint at the basin level from 2008 to 2020. The basin also bears the impact of climate change, especially in regions with limited water resources (*Rani et al., 2022*). The agriculture sector is the primary user of water in the basin. Thus, a more comprehensive approach was taken to assess the WF of major crops in the basin. Sixteen major crops cultivated in the basin were selected for the study based on their total cultivated and irrigated area. They account for 94.0% of the total cultivated and 89.6% of irrigated area annually.

## Methodology

The water footprint was estimated using the AquaCrop model spatially over the study period following the Water Footprint Network guidelines (*Hoekstra et al., 2011*). AquaCrop is a robust crop water productivity model developed by FAO's land and water division. It simulates soil water balance, crop growth, and yield response to water using a relatively small number of explicit and mostly-intuitive parameters and input variables. This model was calibrated and validated for various crops under different conditions. It has been utilized for determining WF at different levels (field scale, basin, and regional).

The AquaCrop model requires the daily rainfall, minimum and maximum temperatures, reference evapotranspiration (ET$_o$), and the mean annual atmospheric carbon dioxide concentration as input climatic data to run (*Steduto et al., 2009*). Daily gridded datasets of precipitation and temperature for the study period were obtained from
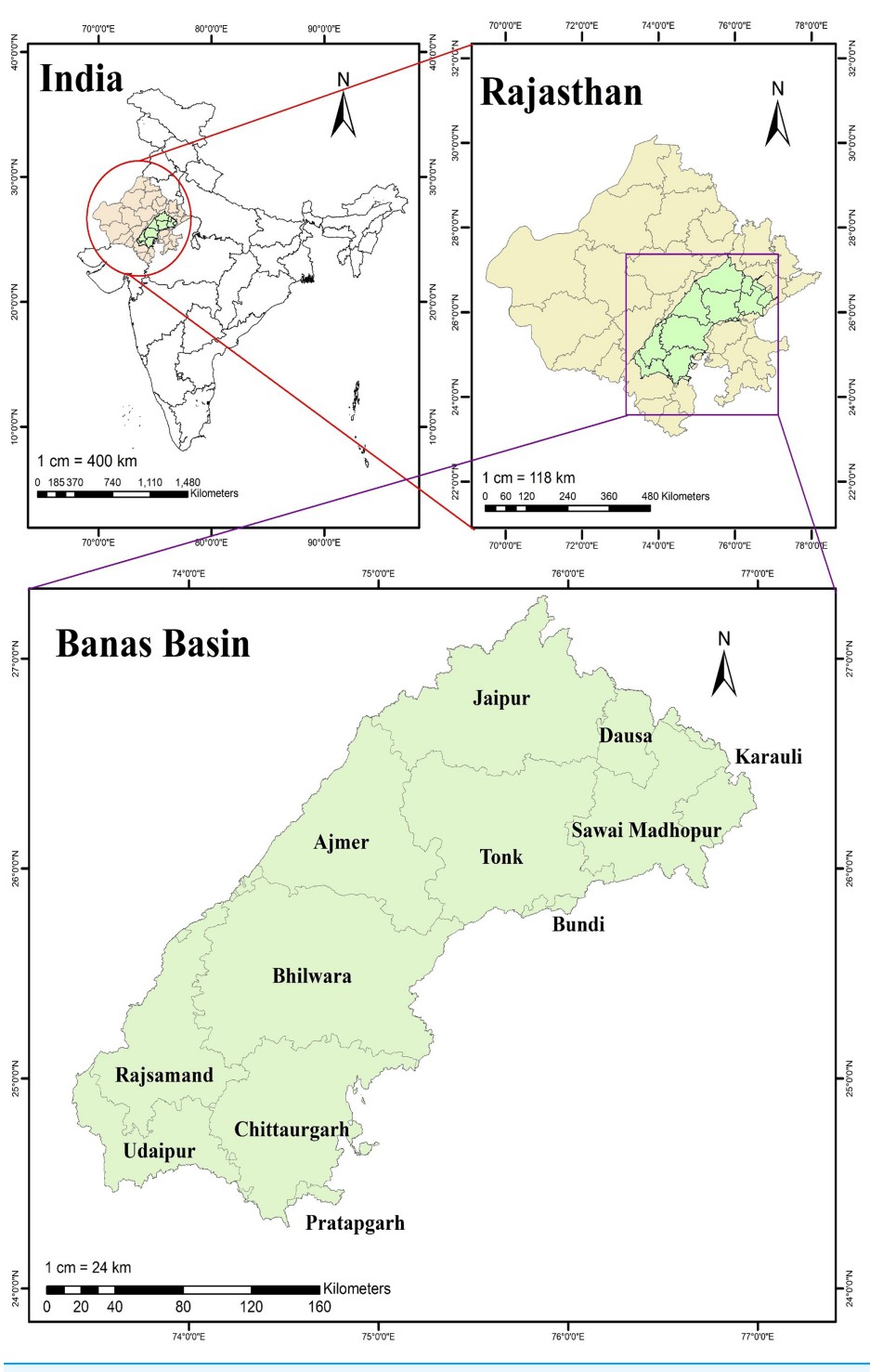

**Figure 1 Location of Banas river basin.**

the India Meteorological Department (IMD). Daily wind speed, relative humidity, and solar radiation data were obtained from the Modern Era Retrospective-Analysis for Research and Applications (MERRA-2) assimilation model dataset. Daily gridded datasets are rescaled to 0.5° × 0.5° spatial resolution to keep consistency. After quality checks and

**Table 1 Datasets used and their sources.**

| S. No | Type of data | Source |
|---|---|---|
| 1. | Shuttle Radar Topography Mission Digital Elevation Model (SRTM DEM) | SRTM DEM, National Aeronautics and Space Administration (https://earthexplorer.usgs.gov/). |
| 2. | Agro-ecological regions map | National Bureau of Soil Survey & Land Use Planning, Indian Council of Agricultural Research (http://geoportal.icar.gov.in/) |
| 3. | Soil properties | Harmonised world soil database v1.2 (http://www.fao.org/) |
| 4. | Land use land cover map | Bhuvan, National Remote Sensing Centre, Indian Space Research Organisation (https://bhuvan.nrsc.gov.in/) |
| 5. | District-wise cropped area and agriculture statistics | Agriculture Statistics Handbook, Directorate of Economics & Statistics, Department of Planning, Government of Rajasthan (https://agriculture.rajasthan.gov.in/) and Agriculture Statistics at Glance, Minister of Agriculture & Farmers Welfare, Government of India (https://agricoop.nic.in/) |
| 6. | Metrological data | India Meteorological Department (IMD), Ministry of Earth Sciences, Government of India (GOI) (http://www.imdpune.gov.in/) and Modern-Era Retrospective analysis for Research and Applications, Version 2 (MERRA-2), NASA (https://power.larc.nasa.gov/) |

processing, daily $ET_o$ was calculated using the FAO Penman-Monteith equation. Major data used in this study and their sources are given in Table 1. The basin area was divided into homogenous land units based on land use, soil, and agro-climatological characteristic to account for spatial variations while reducing the number of simulations required (*Mali et al., 2015*, *2019*). Different thematic layers, namely soil, AESR, LULC, basin boundary, and district boundaries, were overlaid, and LU polygons were formed for each district using intersect feature in ArcGIS.

The plug-in version of the AquaCrop model was used in this study to assess crop WF over the basin because of its flexibility and ease of use for multiple simulations (*Raes et al., 2018*). For simulating various crops, parameterization and calibration guidelines provided by the FAO were followed (*Steduto et al., 2012*). As per their recommendation, crop parameters derived from the available literature were used for the first simulations, and outputs were compared with observed values, then adjusting the parameters and rerunning the simulation. This approach was repeated until the simulation findings roughly matched the observed data. The initial simulation parameters were derived from the AquaCrop user manual (*Raes et al., 2018*). Water fluxes are divided into a crop's green and blue water footprint by following the post-processing of soil water balances (*Chukalla, Krol & Hoekstra, 2015*). Grey water footprint and leaching runoff fractions were determined using the Tier-1 approach recommended by WFN (*Hoekstra et al., 2011*; *Franke, Boyacioglu & Hoekstra, 2013*).

Green and blue WF were obtained by dividing the respective crop water use (CWU) with the yield (Y) over the season.

$$WF_{green} = \frac{CWU_{green}}{y} \tag{1}$$

$$WF_{blue} = \frac{CWU_{blue}}{y} \tag{2}$$

where,

CWU$_{green}$: Green water consumption (m$^3$)
CWU$_{blue}$: Blue water consumption (m$^3$)
WF$_{green}$: Green WF (m$^3$/ton)
WF$_{blue}$: Blue WF (m$^3$/ton)
Y: Yield (ton)

The grey water footprint (WF$_{grey}$, m$^3$/ton) refers to the quantity of water required to assimilate pollutants load as per the ambient water quality standards (generally refers to the maximum and permissible water quality standards). It is given by the equation,

$$WF_{grey} = \frac{(\propto \times AR)/(c_{max} - c_{nat})}{Y} \tag{3}$$

where,

AR: application rate of fertilizers to the field per hectare (kg/ha)
$\propto$: leaching runoff fraction (%)
c$_{max}$: maximum acceptable concentration (kg/m$^3$)
c$_{nat}$: natural concentration for the pollutant (kg/m$^3$)
Y: crop yield (ton/ha)

The water footprint of crop production (blue, green, and grey) was estimated by multiplying the crop WF with the production statistics of the crop and is presented as million cubic meters per year. Besides agriculture, other sectors are equally crucial for the development and sustenance of humankind. We adopted the WF of domestic, livestock, energy, wildlife, forests, and industries sectors from the district-wise water demand of various sectors (WRD, 2014b). This data was developed by adopting standard procedures and local datasets using the Water Evaluation and Planning (WEAP) model. Water demand and availability are intended to be included in a useful tool for water resource planning by using the WEAP system. WEAP stands out for its flexible strategy and integrated approach to modelling water systems. The WEAP puts the supply side of the equation—streamflow, groundwater, reservoirs, and water transfers on an even footing with the demand side, which includes diverse water consumption and its patterns, equipment efficiency, and allocation. District-wise WFs of domestic, livestock, energy, wildlife, forests and industries sectors were estimated using simple linear interpolation for the study period and distributed proportionately based on the area of a district within the basin. These estimates are based on the data of the census population of 1961, 1971, 1981, 1991, 2001, and 2011 future population has been projected up to the year 2060 for the state as a whole for the total, rural and urban population. The district-wise population projections have been made by the ratio method. For the livestock sector, the available data from the livestock census was used. Further details of the methodology used for water demand estimation for other sectors can be obtained from the report (WRD, 2014b). The methodology of WF assessment at the basin scale is illustrated in Fig. 2.

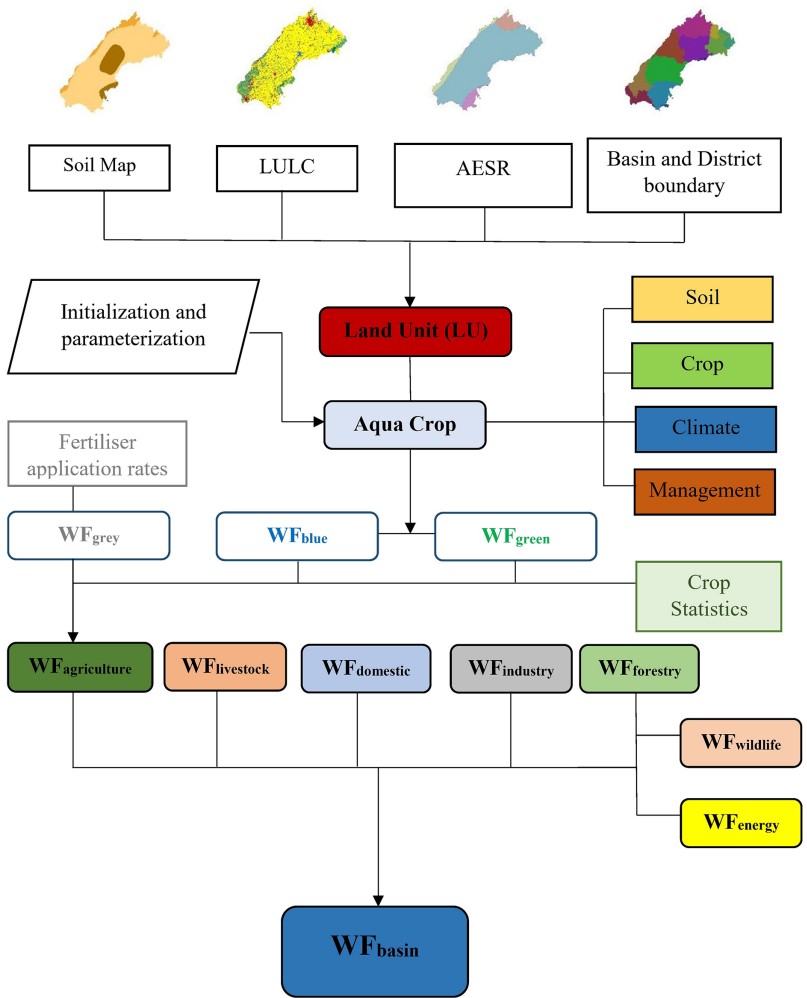

**Figure 2 Water footprint assessment methodology at basin scale.**

## RESULTS

### Water footprint of crop production

The WF was multiplied with crop statistics to estimate WFs of crop production in million cubic meters (MCM) per year. The total annual WF of major crops in the basin was 19,254.5 MCM/year. Wheat, bajra, maize, and rapeseed & mustard make up 67.4% of the total average annual WF of crop production in the Banas Basin (20.2%, 18.3%, 15.8%, and 13.1%, respectively). The annual blue WF of crop production was 3,942.1 (MCM/year). Wheat, and rapeseed & mustard make up almost 87.0% of the average annual blue WF (66.7% and 20.3%, respectively). The largest total WF in the basin was found in wheat (3,890.5 MCM/year), followed by bajra (3,532.7 MCM/year), and then maize (3,040.5 MCM/year). Green WF was highest in bajra (3,213.5 MCM/year), maize (2,776.1 MCM/year), and rapeseed & mustard (1,371.2 MCM/year). Blue WF of wheat was highest (2,629.8 MCM/year), followed by rapeseed & mustard (799.9 MCM/year) and barley
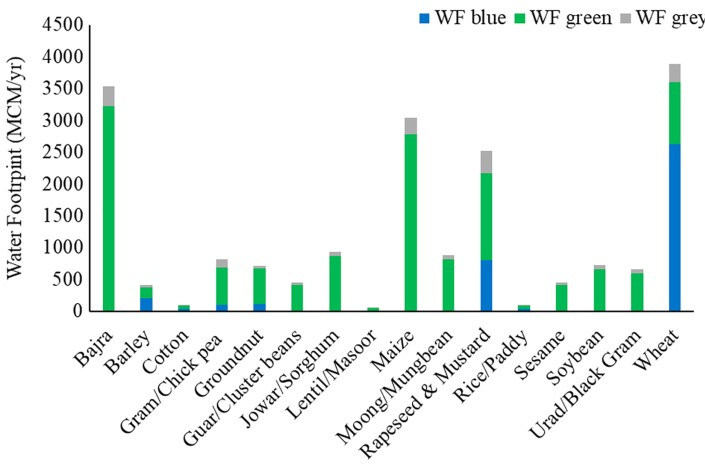

**Figure 3  Average annual water footprint of major crops in basin.**

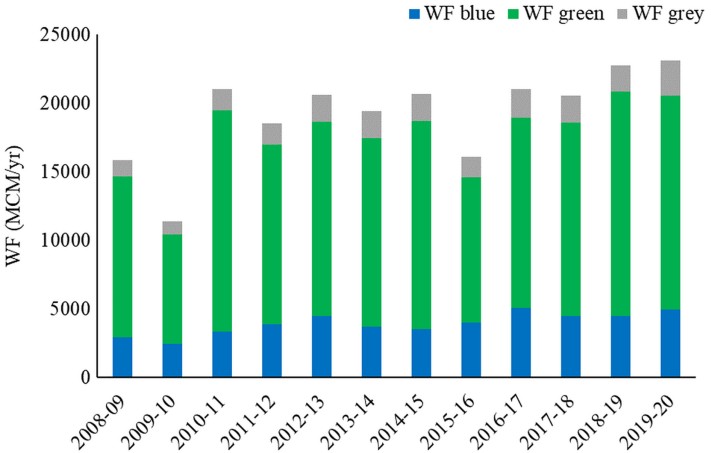

**Figure 4  Average annual water footprint of crop production during 2008–2020.**

(209.8 MCM/year). The largest grey WF was seen in rapeseed & mustard (348.0 MCM/year), bajra (306.2 MCM/year), and wheat (295.5 MCM/year), respectively. Large WF is directly linked with the crop's average WF and the crop's production in the basin. The crop with high production has higher WF in general. The average annual WF of major crops produced in the Banas basin is shown in Fig. 3.

The average annual WF of crop production during the study period is depicted in Fig. 4. The total WF for crop production was found to be highest at 23,131.5 MCM/year in 2019–2020 and the lowest at 11,365.8 MCM/year in 2009–2010, respectively. Spatial variation of blue, green, grey, and total WF of agriculture production for major crops in the Banas Basin is presented in Fig. 5. The blue WF of crop production varies between 82.2–668.5 MCM/year (mean 328.5 MCM/year) in the districts of the basin. Similarly, green WF ranges between 232.3–2,625.5 MCM/year (mean 1,129.9 MCM/year) in the basin districts. Grey WF of crop production varies between 30.8–303.8 MCM/year (mean

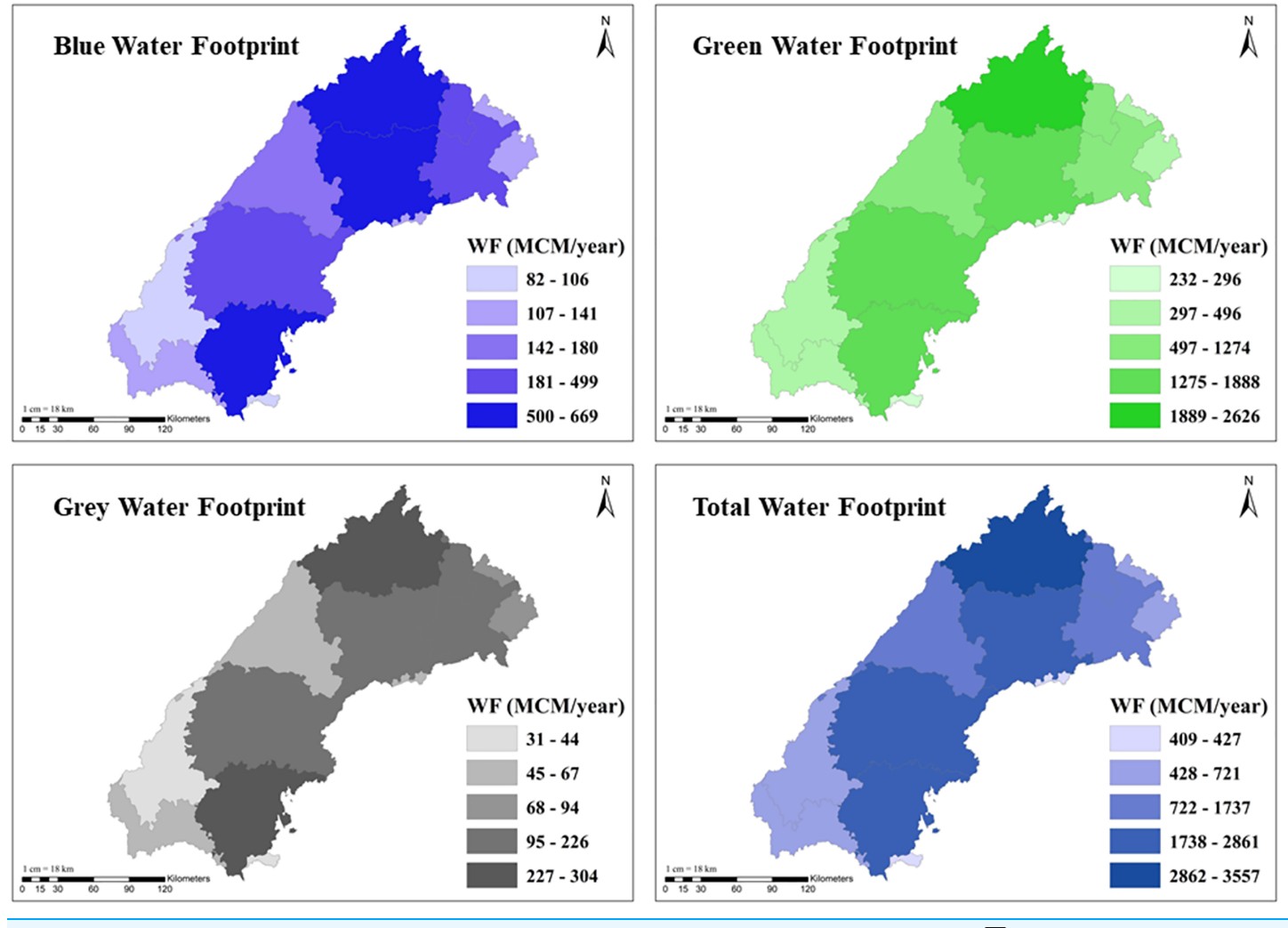

**Figure 5  Spatial variation of blue, green, grey and total water footprint in the Banas River Basin.**

146.1 MCM/year). The highest total WF in the basin was seen in the Jaipur district (3,557.1 MCM/year), followed by Chittaurgarh (2,860.6 MCM/year). The lowest total WF in the basin was found in Pratapgarh (408.6 MCM/year), followed by the Bundi district (427.1 MCM/year). The WF of agriculture is directly linked with crop production, cultivated area, and yield. Hence, districts with a smaller area in the basin have lower annual WF.

## Water footprint of Banas river basin

The water footprint of domestic, livestock, energy, wildlife, forests, and industries sector were derived from the district-wise water demand of various sectors from results from the WEAP model from a study conducted by the Water Resource Department, Rajasthan (*WRD, 2014b*). District-wise water demand data for various sectors from this report was interpolated using simple linear interpolation for the study period. District-wise, WFs were distributed proportionately based on the area of the district within the basin. The total WF of the Banas River Basin from all sectors was 20,238.3 MCM/year. The average annual WF in the various sector was in the following order: Agriculture (19,254.5 MCM/year),

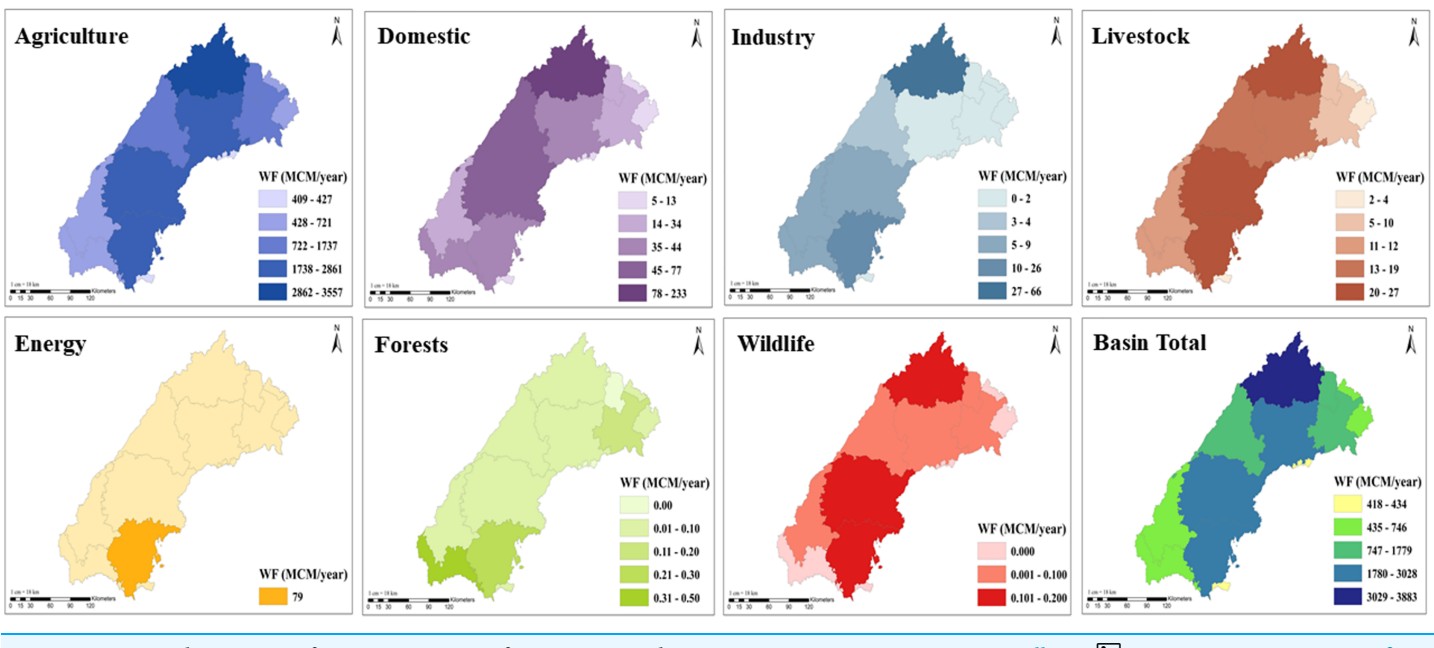

**Figure 6** Spatial variation of sector-wise water footprint over the Banas River Basin.

Domestic (631.4 MCM/year), Livestock (146.8 MCM/year), Industries (123.7 MCM/year), Energy (79.1 MCM/year), Forests (1.7 MCM/year) and Wildlife (1.1 MCM/year). The spatial variation of sector-wise WF over the Banas river basin is presented in Fig. 6.

The agriculture sector accounted for nearly 95.5% total WF of the Banas Basin, which was followed by the Domestic (3.0%), Livestock (0.8%), and Industry (0.5%) sectors, respectively. WF in the Banas Basin was found to be highest at 24,337.5 MCM/year in 2019–2020 and the lowest at 12,167.7 MCM/year in 2009–2010, respectively. WF has increased over the year with the increase in population, rise of industries, and increased demand for crop production in the basin region. Sector-wise, the WF during the study period is shown in Fig. 7.

## DISCUSSION

Among the various crops highest total WF was found in sesame, followed by urad and moong under both irrigated (16,203.6, 11,892.1, and 11,043.9 m$^3$/ton, respectively) and rainfed conditions (14,261.4, 10,359.1 and 9,655.1 m$^3$/ton, respectively). WF is directly proportional to crop water use (CWU) and inversely proportional to crop yield. The average productivity of these three crops was among the lowest and is the major reason for high WF. CWU in rainfed crops was lower in comparison with the irrigated crop. Total WF was found lowest in barley, followed by wheat, then rapeseed & mustard under both irrigated (1,498.6, 1,824.1, and 3,200.6 m$^3$/ton, respectively) and rainfed conditions (1,241.3, 1,508.3, and 2,465.4 m$^3$/ton, respectively). These crops had a higher yield which could be the main factor in the lower WF. It should be noted that higher or lower WF does not mean higher or lower water use per hectare. Most crops have a lower WF under rainfed conditions mainly because crop yields do not necessarily decrease directly with water stress, as the duration and timing of water stress is also a critical factor.

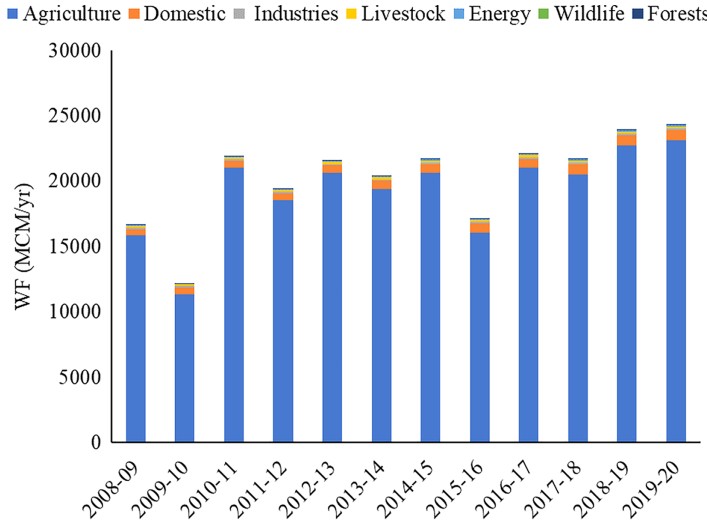

**Figure 7 Sector-wise water footprint during the study period.**

Also, rainfed agriculture is largely practiced in the *kharif* season, where rainfall is plentiful. Significant spatial and temporal variation was seen in WFs in the basin over the study period. For example, the overall WF of the wheat crop under irrigated condition varied between 1,682.8–2,133.2 m³/ton (mean 1,824.1 m³/ton) over the basin. Spatial variation of blue WF during the study period ranged between 1,092.6–1,451.2 m³/ton (mean 1,242.7 m³/ton). Blue WF was 68.1% of total WF on average. Average green WF varied in the range of 407.8–510.4 m³/ton (mean 451.1 m³/ton) during the simulation period. Similarly, grey WF varied between 44.5–213.1 m³/ton (mean 130.4 m³/ton). Similarly, the total WF of the wheat crop under rainfed condition varied between 1,336.5–1,716.8 m³/ton (mean 1,508.3 m³/ton) over the basin on average during the 2008–2020 period. Spatial variation of green WF varied in the range of 1,227.1–1,529.5 m³/ton (mean 1,361.3 m³/ton) during the simulation period. Similarly, grey WF varied between 47.1–247.7 m³/ton (mean 147.0 m³/ton).

WF of major crops in the Banas Basin under irrigated and rainfed conditions is presented in Figs. 8 and 9.

A comparison between the outcomes of this study and earlier research work is given in Table 2. Our study results are in line with previous studies. In the present study, the AquaCrop model was used to estimate WF spatially over time using local data. The reference evapotranspiration was calculated according to the Penman-Monteith equation, which is the most widely used technique (*Allen et al., 1998*). The WF of most crops in the Banas Basin was higher in comparison with the global averages (*Mekonnen & Hoekstra, 2011*). This is basically due to lower yield and climatic variation. Several WF studies have been conducted on different crops at different spatial scales and geographical locations. Only a few studies have been conducted in India, and most use global or national statistics (*Kampman, Hoekstra & Krol, 2008*; *Suhail, 2017*). Then there are global WF studies of crops and derived crop products which also include India

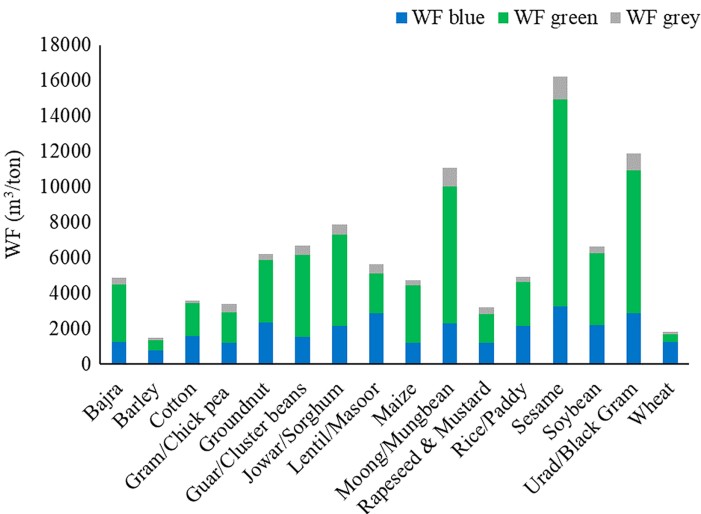

**Figure 8 Water footprints of major crops in Banas River Basin under irrigated condition.**

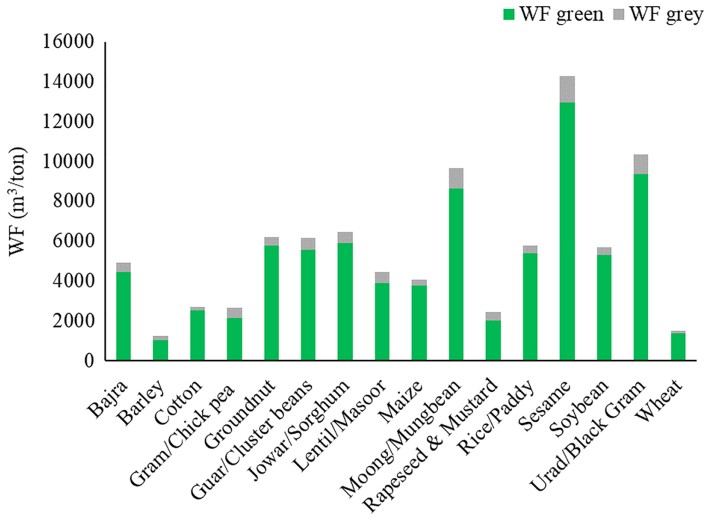

**Figure 9 Water footprints of major crops in Banas River Basin under rainfed condition.**

(*Chapagain & Hoekstra, 2008*; *Mekonnen & Hoekstra, 2011*). Previous basin-level studies conducted in India used simple computation methods using the CROPWAT model (*Mali et al., 2018*; *Rao, Hardaha & Vora, 2019*). Some recent studies have used the AquaCrop model in different regions/basins of the world for WF assessment (*Zhuo & Hoekstra, 2017*; *Nouri et al., 2019*; *Khan et al., 2021*). The primary reason for differences in computed WFs could be the variances in the methodology adopted, the technique used for ET estimation, input data, the model used, the scale, and the scope of the studies. Best efforts were made to parameterize and run the model using locally available data to capture the variation of water footprint adequately. We note that AquaCrop has inherent limitations in modelling crop yield spatially (*Chukalla, Krol & Hoekstra, 2015*; *Berhane, 2018*). A modified default

**Table 2 Comparison of present results with previous work.**

| | Chapagain & Hoekstra (2008) | Kampman, Hoekstra & Krol (2008) | Mali et al. (2019) | Rao, Hardaha & Vora (2019) | Suhail (2017) | Mekonnen & Hoekstra (2011) | Current study Irrigated | Current study Rainfed |
|---|---|---|---|---|---|---|---|---|
| Bajra | 3,269 | 4,222 | | | 4,029 | 4,478 | 4,854 | 4,908 |
| Barley | | | | | 2,124 | 1,423 | 1,499 | 1,241 |
| Cotton | 8,264 | 10,633 | | | | 4,029 | 3,584 | 2,713 |
| Gram/Chickpea | 2,712 | 2,071 | | 9,663 | | 4,177 | 3,382 | 2,649 |
| Groundnut | 3,420 | 4,372 | | 4,085 | | 2,782 | 6,213 | 6,205 |
| Guar | | | | | | | 6,699 | 6,148 |
| Jowar/Sorghum | 4,053 | 3,589 | | 3,739 | 6,026 | 3,048 | 7,855 | 6,463 |
| Lentil/Masoor | | | 5,860 | | | 5,874 | 5,626 | 4,432 |
| Maize | 1,937 | 2,399 | 1,818 | 2,886 | 2,537 | 1,222 | 4,717 | 4,066 |
| Moong/Mungbean | | | | | | | 11,044 | 9,655 |
| Rapeseed & Mustard | 2,618 | 3,972 | | | | 2,809 | 3,201 | 2,465 |
| Rice/Paddy | 4,113 | 4,073 | | 7,848 | 2,070 | 1,673 | 4,897 | 5,767 |
| Sesame | | | 8,956 | | | 9,371 | 16,204 | 14,261 |
| Soybean | 4,124 | 3,526 | | 3,060 | 4,410 | 2,145 | 6,635 | 5,711 |
| Urad/Black Gram | | | | | | | 11,892 | 10,359 |
| Wheat | 1,654 | 1,412 | 2,473 | 5,417 | 2,100 | 1,828 | 1,824 | 1,508 |
| Study Period | 1997–2001 | 1997–2001 | 2011 | 2000–2013 | 1999–2006 | 1996–2005 | 2008–2020 | |
| Scale | Global | National | Regional | Regional | National | Global | Regional | |
| Location | India | India | Gomati Basin | Banjar Watershed | India | Global | Banas Basin, Rajasthan | |
| Method | CROPWAT | CROPWAT | CROPWAT | CROPWAT | CROPWAT | CROPWAT | AquaCrop | |

crop file was used to simulate crops when the standard crop file was unavailable in AquaCrop. Still, these results can provide a valuable reference for similar future studies.

On average, the WF of crop production was 69.7% green, 20.8% blue, and 9.5% grey in the basin. Rainfed agriculture is prominent in the Banas river basin and is the reason for higher green WF. In general, the WF of crop production is increasing as more area comes under cultivation of crops, high-yielding varieties of crops are being developed, improved irrigation technologies become available, and more water storage structures are being constructed. These results are in line with previous results from similar studies. The blue WF accounted for 47.3% and 43.6% of the total WF of Gomti and Betwa basins, respectively, while the share of grey WF was about 9.1% and 10.9% of total WF (*Mali et al., 2018*, *2019*). Studies have shown that 78% of the global agricultural WF is green, 12% is blue, and 10% is grey WF (*Mekonnen & Hoekstra, 2013*).

The agriculture sector accounted for nearly 95.5% total WF of the Banas Basin, which was followed by the Domestic (3.0%), Livestock (0.8%), and Industry (0.5%) sectors, respectively. This is similar to one study from India, where crop production accounted for nearly 95.5% and 96.4% of the WF in the Gomti and Betwa basins, respectively (*Mali et al., 2017*, *2018*, *2019*). In China, a study estimated the WF of the Yellow River Basin to be 1768 MCM, 96% of which was from agriculture (92% for crop production and 4% for livestock)

and the rest 4% from industrial and domestic sectors, respectively (*Zeng et al., 2012*). Crop statistics, population, livestock, and water demand data for other sectors are not available at the river basin level. So, we had to calculate it based on district-level estimates and the area of districts within the basin. This inherent limitation leads to errors in the calculation as statistics within the district are assumed to be distributed equally, which may not be accurate in most cases. Many other previous studies on data availability or planning of resources are done on administrative scales instead of the basin. For proper management of water resources, there is a need to implement basin-scale planning and databases. While more focus was put on the agriculture sector for this WF assessment as it is the primary consumer of water in the basin. Evaluation of water demands of other sectors was made based on data reported by the water resource department which was comprehensive but somewhat outdated, and the distinction between the blue, green, and grey components of WF couldn't be made for them. Future studies on various components of WF for other sectors will also be instrumental. Even with their minor contribution to overall WF in the basin, other sectors may significantly contribute blue and grey degenerative WF, which can be crucial for sustainable water use planning.

Agriculture makes up a considerable part of the basin water footprint, and it is necessary to reduce it to sustainable levels. Numerous studies have concluded that WF can be reduced by adopting strategies, methods, and technologies to reduce non-beneficial consumptive water use (*Jovanovic et al., 2020*). Some practices can upgrade the water management in agricultural fields by implementing precision irrigation methods (*Smith, 2011*; *Abioye et al., 2020*), improving irrigation efficiency (*Evans & Sadler, 2008*; *Greenwood et al., 2010*), and irrigation scheduling (*Hinton Consulting, 2001*; *Tesema et al., 2011*; *Wen, Shang & Yang, 2017*), adopting better agricultural practices like drip irrigation and mulching (*Chukalla, Krol & Hoekstra, 2015*; *Nouri et al., 2019*; *Scardigno, 2020*; *Ding et al., 2021*) and augmenting water productivity (*Igbadun, Ramalan & Oiganji, 2012*; *Muhammad, Zhu & Bazai, 2017*; *Mubvuma, Ogola & Mhizha, 2021*). Agronomics practices and *in-situ* water conservation can significantly reduce local water scarcity (*Kumar et al., 2021*; *Sharma et al., 2021*; *Singh et al., 2021*). Reducing food wastage (*Sun et al., 2018*; *Kashyap & Agarwal, 2020*) and focusing on changing diets (*Harris et al., 2017*; *Green et al., 2018*) can also help decrease water consumption.

## CONCLUSIONS

This study provides a comprehensive estimate of the water footprint of various sectors. The water footprint of major crops was estimated using the AquaCrop model spatially over the study period (2008–2020). The water footprint of crop production (blue, green, and grey) was estimated by multiplying the crop water footprint with district-wise production statistics. The water footprint of domestic, livestock, energy, wildlife, forests, and industries sector were derived from the district-wise water demand of various sectors. The water footprint of crop production in the basin was 19.3 BCM/year. Wheat, bajra, maize, and rapeseed & mustard make up 67.4% of crop production's total average annual water footprint. The larger water footprint is directly linked to the cultivated area and production of the crop in the basin. Water footprint of the Banas River Basin was estimated as

20.2 BCM/year from all sectors. The agriculture sector accounted for nearly 95.5% total water footprint of the Banas Basin. The water footprint has increased over the year with the increase in population, the number of industries, and crop production demand. The results of this study provide helpful insights into the current situation in the basin. Appropriate measures are required to develop adaptation approaches to overcome water scarcity challenges in the basin. Outcomes provide baseline information for further research to advance sustainable production and planning. Suitable actions should be taken for improving water productivity and promoting sustainable water use. There is a need to promote practices like changing crop patterns, mulching, and micro irrigation to reduce water use in agriculture.

## INDEX OF NOTATIONS AND ABBREVIATIONS

| | |
|---|---|
| % | Percentage |
| $\alpha$ | Leaching runoff fraction |
| $\approx$ | Approximately equal to |
| $^\circ$ | Degree |
| AESR | Agro-ecological sub region |
| APSIM | Agricultural Production Systems Simulator |
| BCM | Billion cubic meters |
| BRB | Banas River Basin |
| CWU | Crop water use |
| DSSAT | Decision Support System for Agrotechnology Transfer |
| ET | Evapo-transpiration |
| $ET_o$ | Reference evapo-transpiration |
| GOI | Government of India |
| Ha | Hectare |
| IMD | India Meteorological Department |
| LU | Land units |
| LULC | Land use land cover |
| MCM | Million cubic meters |
| Mha | Million hectares |
| PIB | Press Information Bureau |
| WEAP | Water Evaluation and Planning |
| WF | Water footprint |
| WFA | Water footprint assessment |
| $WF_{blue}$ | Blue water footprint |
| $WF_{green}$ | Green water footprint |
| $WF_{grey}$ | Grey water footprint |
| WFN | Water Footprint Network |
| WOFOST | World Food Studies |
| WRD | Water Resource Department |
| Yr | Year |

## ACKNOWLEDGEMENTS

The author expresses gratitude towards all faculty in the Department of Soil and Water Engineering, College of Technology and Engineering, Maharana Pratap University of Agriculture & Technology and Indian Council of Agricultural Research for all their support during the research.

### Funding

The author received no funding for this work.

### Competing Interests

The author declares that they have no competing interests.

### Author Contributions

- Mukesh Kumar Mehla conceived and designed the experiments, performed the experiments, analyzed the data, prepared figures and/or tables, authored or reviewed drafts of the article, and approved the final draft.

### Data Availability

The raw data is available in the Supplemental Files.

### Supplemental Information

Supplemental information for this article can be found online at http://dx.doi.org/10.7717/peerj.14207#supplemental-information.

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
