# Peer review of "Regional water footprint assessment for a semi-arid basin in India"

_PeerJ, doi:10.7717/peerj.14207_

## Round 0.1 · original submission · Major Revisions

The work is interesting, has a clear degree of originality, and is appropriate for publication in the journal after performing a major and very careful revision. Nevertheless, it needs some further improvements. In general, there are still some occasional grammar errors throughout the manuscript, especially the article "the," "a," and "an" are missing in many places; please make spellchecking in addition to these minor issues. The reviewers has listed some specific comments that might help the authors further enhance the manuscript's quality.

In addition:

1. Specific Comments
• You have used many abbreviations in the text. From this perspective, an Index of Notations and Abbreviations would be beneficial for a better understanding of the proposed work. Furthermore, please check carefully if all the abbreviations and notations considered in work are explained for the first time when they are used, even if these are considered trivial by the authors. The paper should be accessible to a wide audience. Furthermore, it will make sense to include also the notations in this index.
• The abstract should be redesigned. You should avoid using acronyms in the abstract and insert the work's main conclusion.
• The objectives should be more explicitly stated.
• What is the novelty of this work?
• It is better to improve your contributions which are not so clear to show the advantage of
• your work.
• The methodology limitation should be mentioned.
Many equations are presented in the paper, and most look OK. However, please check carefully whether all equations are necessary and whether the quantities involved are properly explained. Also, some equations need references.

• Results
• This section is well written.

• Discussion
• Overall, the discussion part is weak. The Discussion should summarize the manuscript's main finding(s) in the context of the broader scientific literature and address any study limitations or results that conflict with other published work.

·

Basic reporting

Literature review should be enhanced and the noverty is very weak.

Experimental design

Method should be elaborated, providing more details.

Validity of the findings

Conlusion should be revised.

Additional comments

This paper has assessed the regional water footprint (WF) in a semi-arid basin of India. This is a traditional topic, although the author uses water footprint assessment in a new case. I suggest the editor reconsider this paper after major revision. My comments are as follows.

1. The introduction is organized with bad logic. Please reorganize this part, emphasizing the significance and necessity of your work.
2. The research question is not clear and the innovation is totally weak. Copious literature involves the WF assessment across the globe including Indian context, so what is the new thing of your work? You should re-review the preexisting literature and critically discuss your contribution. This is my major concern.
3. Line 97-98, you use the area of district as the proportionality factor. Why not use the number of population?
4. For materials and method, you introduce so many datasets, please create a table to summarize such information so that the reader can easily follow.
5. AquaCrop model and WEAP model are your key methods, please briefly mention them, providing more information.
6. There are some minor language issues/typo, like line 194—corps; line 211, please carefully check the full text.
7. Line 200, what does ET0 mean, it should be explained when the variable first occurs.
8. In discussion, the spatial variation of WF in your research area should be further explained. Now, the discussion is insufficient.
9. Research limitation should be briefly mentioned.
10. In conclusion, the main findings include too many figures. The reader may feel dazzled especially for those nonspecialists. Please conclude your most interesting findings.
11. Please enlarge the legend in Figures 5 and 6.

·

Basic reporting

Dear author, I have proceeded to review the manuscript entitled "Regional water footprint assessment for semi-arid basin in India" whose aims was to provide a comprehensive evaluation of the sector-wise water footprint in the Banas river basin for a period from 2008-2020, as well as its philosophical values. Prior to further processing if the editor deems so; it is necessary to cover the following comments very thoroughly:


- The abstract is well structured.
- Some words in the title are repeated in the keywords, please try not to repeat the words in the title. The idea is that, if your manuscript is published, you will have more chance that your article will be read.
- In the final part of the last paragraph of your manuscript when you narrate the objective it seems to me that it appears out of nowhere, try to connect to the previous sentences when you narrate the problem.
- The materials and methods section clearly divides 2.1 area of study and 2.2 methodology.

Experimental design

Regarding the experimental design

- Section materials and methods.
- The figure of the study area is very well presented.
- Try to make clear the data section as you got it, but do not mix it with the model for the water footprint calculation. Also, try to briefly describe the model steps and formulas.
- You need to show in tables or graphs the data obtained from the water footprint.
- Reading the objective I get the idea that you will only analyze in the agricultural sector. But the results and in general the document describes the water footprint consumption from different aspects (agricultural, livestock, domestic, forest...). I think you should reformulate the objective.
- In the discussion I note that you only discuss the agricultural issue. Please discuss the total of your results found.
- In the conclusion if you refer to all the results. Considering that you generated the water footprint with respect to demand, I ask, did you take demographic data here, if so don't you describe it... from what year, did you generate population indices...?
-

Validity of the findings

no comment

Additional comments

The manuscript is very clear. I believe that if it is worked on a little more it could be published, if the editor has no other criteria.

---

## Round 0.2 · accepted · Accept

I congratulate the authors for the effort put into this paper! The manuscript is significantly improved; therefore, I recommend accepting it in its current form!

·

Basic reporting

I have reviewed the updated manuscript and the author’s responses. I feel the authors have well addressed the issues and challenges during my last review. I have no additional suggestions for revision.

Experimental design

No comments

Validity of the findings

No comments

Additional comments

No comments

·

Basic reporting

I am very pleased and satisfied with the work done by the authors. The receptivity can be seen and this has greatly improved the manuscript.

Experimental design

Now the design and its methodology is clear and better understood.

Validity of the findings

no comment